# Caregivers’ Experience of Caring for a Family Member with Alzheimer’s Disease: A Content Analysis of Longitudinal Social Media Communication

**DOI:** 10.3390/ijerph17124412

**Published:** 2020-06-19

**Authors:** Pavel Bachmann

**Affiliations:** Department of Management, Faculty of Informatics and Management, University of Hradec Králové, 50003 Hradec Králové, Czech Republic; pavel.bachmann@uhk.cz; Tel.: +420-493-332-378

**Keywords:** Alzheimer’s disease, caregiver, online support group, Facebook, social media

## Abstract

*Background:* The population aging together with an increased incidence of Alzheimer’s disease (AD) should also be accompanied by a growing interest in healthcare research. Therefore, this study examines the nature of the caregiver’s work, its mental and physical demands, experience and questions, and the relationship between the person with AD, the caregiver, and family members. *Methods:* As social media has become the place where people share family situations, a Facebook private discussion group of caregivers was chosen as the analytical data source. The study documented the daily-life situations of one-hundred dyads based on 2110 posts published during a six-month or longer period. A content analysis classified communication into 35 categories of basic, instrumental, and extended activities of daily livings (ADLs) and newly designed caregiver’s daily issues (CDIs). *Results:* The frequently discussed topics were related to exhaustion and feelings of “giving up” by caregivers and interpersonal communication and help from family members. The highest support was found for the topics of aging and dying and family events. *Conclusion*: The communications of caregivers were diverse and rather associated with co-occupational ADLs and CDIs than basic or instrumental ADLs. The support of the group was mainly provided in coping with fundamental life changes.

## 1. Introduction

Aging is one of the most significant phenomena of the contemporary world, bringing not only changes in family social relations, but also in healthcare services and public, local, and family economies. Globally, the number of persons aged 80 or over is projected to triple by 2050. In Europe, 25% of the population is already aged 60 years and over and this proportion should reach 35% in 2050. Population aging is predicted to have a profound effect on societies and on fiscal and especially health care burdens of every country [1] and bringing new challenges for family care [2,3,4]. Increased life expectancy established dementia as a major public health challenge worldwide. Alzheimer’s disease (AD) is the most frequent form of dementia and accounts for 60–80% of people affected by dementia [5,6]. It is a progressive neurodegenerative disease, irreversible and disabling, producing a high socioeconomic burden, even higher than in cases of other diseases [7,8,9].

Essentially, such a demographic transition will require a new global approach emphasizing innovative and preventive institutional and family healthcare and medical needs of the elderly population [9]. Moreover, the social and economic consequences include relationships between caregivers and elderly individuals, food and nutritional concerns, and the maltreatment of elderly individuals, poor knowledge and awareness about the risk factors, psycho-emotional concerns, financial problems, health care costs, etc. In general, we can talk about the impact on the quality of life on both sides: persons with AD as well as their carers [10,11,12].

In parallel with population aging, there is an enormous rise in the use and spread of social media worldwide. These media have become not only a means of interpersonal communication but also reflect our lives, successes and failures, including seeking support and solutions in the field of human health. In 2019, there were 3.725 billion active social media users which was 83.2% of worldwide Internet users and 48% of the worldwide population; this percentage increased by 9.6% from 2018–2019 [13]. Social media also provided new ways to assist persons with AD, nurses and families in gaining support in life and health situations [14].

Moreover, social media platforms not only make it possible to gain support in complex situations, but also enables presentation and documentation of relevant information. Specifically, the Facebook support group used in this study provided a very large database including daily activities and situations, emotions and experience that describe the relationships between the carers and people with AD. For this reason, it might be very fruitful to interconnect such evidence on daily lives with the activities of daily living (ADLs) classification. Therefore, this study aimed at identification of the content associated with the caregiving process and, consequently, verifying the validity of current ADLs concepts. Since the preceding concepts primarily work with a focus on people with AD, the study also concentrates on the classification of new, yet undescribed problems on the part of caregivers and the design of their structure. Particularly, the frequency of communicated content as well as the level of its acceptance by other caregivers is to be examined.

## 2. ADLs, Caregiver Burden and Social Media Support

### 2.1. ADLs and Its Classification

The ADLs classification was first proposed in the 1950s by Sidney Katz and his team [15] and was frequently in the focus of health-related research studies aimed at older adults. Almost two decades later, in the study of Lawton and Brody [16], it was proposed as a list of activities of physical self-maintenance activities and instrumental activities of daily living.

Several concepts of ADLs were developed over the past years. The basic ADLs include fundamental skills typically needed to manage basic physical needs: personal hygiene, dressing, toileting, transferring, and eating [17,18]. The basic ADLs are often classified separately from instrumental activities of daily living (IADLs), which contain complex activities related to independent living, for example, managing money, preparing meals, using a telephone or computer, or taking prescribed medicine [18,19]. Besides the mentioned concepts, there is a view of American occupational therapy association which identified another 12 types of extended activities of daily living (EADLs) that are performed together with other persons; these activities are related to, for example, church observances, celebration of anniversaries, birthdays or other special events; care of pets; or safety procedures and emergency responses [20].

The mentioned concepts are often used in dementia-oriented research, and a significant part of this research is devoted to non-pharmacological therapies and interventions [21,22,23]. Specifically, the studies deal with physical [24,25] and occupational therapies [26,27] or interventions for neuropsychiatric symptoms of dementia [28]. Moreover, the research in this field is frequently associated with the burden of family caregivers associated with the care of people with dementia, particularly with AD [29,30,31,32].

### 2.2. Dementia Caregiver Burden

The risks associated to carers resulting from long-term and usually unpaid care for people with dementia or specifically AD is referred as caregiver burden. Caregiver burden has been defined as a “stress response elicited by negative self-appraisal of coping capacity and/or resources needed to meet demands of caregiving” [33], or as an outcome of the interaction between characteristics of person with dementia/AD and the caregiver, where both of them can act as determinants [34]. Family caregivers are socially isolated and burdened during their long-term care for their loved ones [35].

The research studies particularly deal with the caregiver burden experience by informal caregivers of the person with AD [36,37,38]. A meta-analysis found that dementia family caregivers are significantly more stressed than non-dementia carers and suffer more serious depressive symptoms and physical problems [39], specifically the overall prevalence rate of depressive and anxiety symptoms is 34% and 44%, respectively [40]. The physical and psychological burden of providing ADLs can severely influence not only the family caregivers, but the whole family [41,42]. Caregivers need to limit their time of their self-care and socializing [43] and are exhibited to higher risk of depression and health problems, in general [44,45]. Despite of the fact that persons with AD in the late stages are not often involved in the research studies, the burden is growing with the progression of the disease due to loss of basic and instrumental activities of daily living and specific person’s behaviour as agitation, aggression and disinhibition followed by delusions and mood disturbances [29,30].

### 2.3. Caregiver Support through the Social Media

The healthcare workers, including dementia caregivers, used a portfolio of social media tools, such as social networking sites (e.g., Facebook, Twitter, YouTube, Instagram, or LinkedIn), blogs, and microblogs, wikis, and learning management systems, to share their daily practice. By understanding the social media as the part of the caregiving experience the new interventions and services aimed at improving caregiver burden and quality of life can be developed [46]. Social media affects the healthcare professional/patient relationship, leading to more equal communication and increased switching of doctors [14,47]. It is also worth remarking on the dangers of social media use in healthcare research specified by Ventola [48]: reduced quality of information, posting of unprofessional content, the first impression based on the user’s personality and not on the content, breaching of people’s privacy, violation of the patient-HCP boundary, licensing and legal issues.

The use and effectiveness of social media to support population of caregivers of persons with AD has been mentioned relatively frequently in the literature [47,49,50]. In the case of more common diseases, such as cancer or arthritis, the discussion of support groups is used for the patients themselves [51]; however, in the case of older adults’ diseases, they serve rather as a support for caregivers and family members [52,53]. In general, the health content on social media is rather impersonal and related to marketing [54], and the private discussions on Facebook are more suitable for support delivery to caregivers of people with AD [52,53]. Facebook discussion groups are also often used as a platform for identification of qualitative pictures through focus groups [55]. The systematic review on the effect of Internet-based group support [35] showed a positive effect on social support and self-efficacy of caregivers. More detailed benefits of such support were provided by a study from Lagervall et al. [56], who found that Facebook groups encouraged mental well-being and are particularly advantageous for caregivers of individuals diagnosed with dementia, as it is difficult to leave loved ones with dementia alone.

## 3. Materials and Methods

### 3.1. Content Analysis in Social Media Environment

This study used content analysis of discussions as the main method. The literature definitions stress either a quantitative or a qualitative approach to the analysis with dependence on the discipline or time trend; a combination of these two main approaches was used in this study. The concept of ADLs served as an a priori coding scheme including specific categories for classification. However, during the classification, rather the semantics and prevailing meaning and intent behind the published post content, as is recommended by Weber [57], was investigated. In the case of communication which was not reflecting ADLs but rather structure of caregivers’ emotional and life situations, we applied constant re-contextualizing and re-defining [58,59] of categories and also subjective interpretation of the content of text data through the systematic classification process and identifying themes or patterns [60].

Specifically, an analysis was conducted in the private discussion group environment of the Facebook social networking site. Such research conducted in an online environment is a part of so-called Internet-mediated research [61]; social media are defined as a “group of Internet-based applications that allow the creation and exchange of user generated content” [62]. Social media behaviour can be classified into three levels: “consuming, contributing and creating” [63]. Consuming means the passive and non-contributing behaviour, mainly reading (or watching) other’s posts available in the discussion group. Such behaviour is typical for the highest percentage of a group’s members. Contributing include members who are interacting with published posts in the form of reacting (likes and other reactions) or commenting on them. Such behaviour often involves support directed to the post author. Creating represents the highest involvement in the group and involves production of group contents. According to Piolat et al. [64] people use more cognitive effort when creating (writing) than when consuming a message. Also, the size and nature of issues presented by the discussion depends on the personality of its author (e.g., his life philosophy or life experience) [65]. Since the discussions of persons with AD and caregivers include various life situations, problems and issues, a very broad coding scheme was selected by including three main concepts of ADLs: basic activities, instrumental activities, and extended activities. The content of the concepts mentioned is provided in the Box 1.

Box 1Categories of ADLs concepts used for coding.
(A) Basic ADLs [27](1) Bathing and showering; (2) Personal hygiene (excluding 1); (3) Dressing; (4) Toilet hygiene; (5) Transferring (mobility within the house); (6) Self-feeding, eating/drinking (including chewing and swallowing); (7) Communication and interaction.(B) Instrumental ADLs [28](1) Cleaning and maintaining the house; (2) Managing money; (3) Moving within the community (outside the house/the family); (4) Preparing meals; (5) Shopping; (6) Taking prescribed medications; (7) Using the telephone, computer and online communication; (8) Entertainment (e.g., movies, music, books, pictures, videos); (9) Physical activities, sports.(C) Extended ADLs [29](1) Selection of caregivers and health facilities, moving to a facility; (2) Care of pets; (3) Safety procedures and emergency responses; (4) Sleeping disorders, including sundowning; (5) Religious observances, holidays, birthdays, anniversaries; (6) Health management, maintenance and prevention; (7) Social groups and other public events; (8) Ageing, death and dying; (9) Getting help (especially from family members); (10) Appointments, tests, surgeries.


An analysis investigated the quantitative and qualitative extent of each ADLs concept and its categories. In the quantitative extent, there were two main variables: the frequency of posts published in a category and the engagement of other group members were examined. The frequency of posts was expressed in its absolute and relative amount. The engagement level of group members was based on the current practice of Facebook [66], where this metric includes variables such as number of reactions, number of comments, and number of shares. In this study, as the exact weight assigned to the variables included in this engagement indicator was not known, we worked with a sum of a single number of reactions (likes, and love, haha, wow, sad, and angry reactions) and the single number of comments. To the qualitative extent, the character of each category was described with key discussion topics based on the posts published.

Facebook discussion group was used as data source in this content analysis for several reasons: (1) these type of groups reflect the daily life activities and issues of caregivers and enable to monitor them retrospectively and in the long-term; (2) the research studies mentioned earlier confirmed the positive effect of such group support on the caregivers; (3) this source of data is relatively novel, and thus an analysis can provide not only new findings but also unique ways and techniques on how such data can be analysed.

### 3.2. The Sampling Strategy

Facebook with 2449 million active monthly users was the biggest social network worldwide in 2019. In developed countries, the share of Facebook users exceeds 90% of population [67]. Therefore, this social network was considered an appropriate platform for analysis. Although discussions can take place in various spaces, such as comments published below posts on a page or in community of for a specific page, reviews, and others, the basic tool for peer communication on this network are groups. According to Facebook, groups are designed for group discussions and for people to share their common interests and express opinions.

In order to collect the most representative data source, the following keywords were used to analyse Facebook groups: “Alzheimer”, “AD”, “Alzheimer’s” or “Alzheimers”. Only groups with over 500 members and those which operated in English were considered in the study. In total, 28 groups (20 private and 8 public) met set restrictions. The number of group members ranged from 620–142,000. Due to the preceding verifications in research studies [50,68] and expected higher openness of members and lower frequency of unwanted spam communication, only private (called “closed” in the past) discussion groups were considered in our study.

A Facebook group titled “Alzheimer’s and Dementia Support” was selected for in-depth analysis as the largest private discussion forum with 30,911 members (valid to 17 January 2020). This group is aimed at peer-to-peer support to caregivers of persons with AD, and the membership is restricted to people who know a person affected by a neurodegenerative disease. The group consists of people living mainly in the United States, United Kingdom and Canada. The overall sampling strategy is illustrated in Figure 1.

The initial step of the selection of individuals and the content published by them was to verify whether the group members communicate about care provided to a person with AD repeatedly and over the long term to obtain a multifaceted picture of the caregiving process. It was found that a vast majority of the members are rather passive and publish from zero to two posts per year; members publish 2.3 posts per year on average. To avoid distortion of data by such low-frequent communication, which is usually emotionally strong and does not reflect real and complex life, the longitudinal research approach was used. The members publishing at least ten posts during at least six-months’ time-period were considered and all the posts published by them were collected. In the next step, the content published by the sampled caregivers, but not directly focusing on the relationship with the person with AD (such as funny, educative or fundraising content) was excluded from the analysis.

### 3.3. The Sample Characteristics

The sample included 100 group members who declared themselves as caregivers and published their statements repeatedly and over the long term were selected for the study. In total, 2110 caregiver posts published in between 10 March 2017 and 28 December 2019 were collected. During the monitored period, one caregiver included in the sample published 21 posts on average.

In the sample, the 88 women and 12 men were on the side of caregivers. They identified themselves as family caregivers in following way: more than half of them (52%) declared themselves as caregivers explicitly in the text (often they also declared they are alone in performing this job). One-third of them (33%) were identified as caregivers from the context they published. The remaining 15% considered themselves as minor caregivers (or supporters) of close relatives placed in permanent health care facilities (hospices, nursing homes, etc.). Regarding the family relationship, the structure of caregivers comprised 67 daughters (5 daughters-in-law were included), 16 wives, 6 husbands, 5 sons, 3 grandchildren (2 females and 1 male) and one sister who ensured the caregiving activities (also, in 2 cases, the relationship was not determined). The sampled caregivers formulated their posts as testimony (1454 posts, 68.9%), the testimony combined with question (307 posts, 14.5%); questions (301 posts, 14.3%) or as recommendations (48 posts, 2.3%).

On the side of persons with AD, the 103 people were monitored. The sample comprised 63 mothers (including three mothers-in-law), 16 husbands, 6 wives, 6 fathers and 3 grandmothers. Besides this, in three cases, the care was provided by one caregiver to both parents with AD. Persons with AD were most often in stage 4, 5, or 6 or the stage could not be detected.

### 3.4. Data Collection, Processing and Inter-Coders Reliability

#### 3.4.1. Data Collection

More approaches for data collection on social media exist, for example Zhao, Zhang and Min [68] applied natural language process technique to unstructured health data collection on social media, and Yang, Lee and Kuo [69] used sentiment analysis during the process of social media data-mining. However, in this case, manual coding was used to obtain long-term communication of specifically selected and individual caregivers to capture their experience, doubts, questions, and other communication related to the caregiving process. The selection of appropriate members who met pre-determined research conditions was facilitated by use of a Facebook badge “Conversation Starter” which recognizes active members. The posts were collected manually and coded by two coders, the first of them is the author of this study, and second one is his student. The student was carefully informed before the coding started and the coding scheme was clearly explained. Each post was assigned to just one category according to its main content. Following variables were coded (if available): post id, anonymized name of the post author, gender of the post author, country of the post author, the post content, number of post reactions (including all its types: likes, love, haha, wow, sad, and angry reactions), number of post comments. Next, based on in-depth content analysis, more data were collected: gender of the person with AD, person with AD relationship to the caregiver, the type of the caregiver (main, minor) and the classification into individual categories of ADLs was made. The data were collected retrospectively in the range from six months to three years.

#### 3.4.2. Data Processing

The data were recorded and processed in Microsoft Excel. For protection of personal data, the users’ name data as well as the names of specific places and the names of country states were anonymized. The quantitative aspect of research included statistical processing of two basic variables: frequency of communication and total average engagement. The descriptive statistics was used to pronounce frequency of communication. A one-way analysis of variance (ANOVA) statistics with medium effect size (f = 0.25) and Tukey HSD testing were used to identify differences in total average engagement of discussers in individual ADLs categories.

#### 3.4.3. Inter-Coders Reliability

Inter-coders reliability refers to the extent to which more independent coders agree on the coding of the content applying the same coding construct. As two independent coders were employed in this analysis, the sample of one-tenth of the posts (211 posts) was used to test this reliability, which means that this number of posts was coded by both coders employed in this analysis. A non-high reliability agreement score was expected, as the number of categories was relatively high (35) and the testimonies of caregivers were very complex and usually could be classified in more categories. However, the testing showed only 15 disagreements in coding which resulted in high reliability—over ninety percent (92.9%). Such high agreement score proves that the selection of the coding scheme was appropriate as well as that the coders had a professional expertise in the examined area.

The disagreements in coding were found mainly (in 9 cases out of 15) in testimonies related to the celebration of anniversaries, birthdays, etc. (included as a part of extended instrumental ADLs) where caregivers also expressed fear about the future life (included as a part of caregivers’ issues).

#### 3.4.4. Ethical Aspect of Data Collection and Processing

In relation to the ethical aspect of data collection and use, it is important to note the following. The researcher informed the group administrators about his intent to analyse the daily life of caregivers. At the same time, the data were collected with the use of legal tools designed by the Facebook. Data on individual caregivers were anonymized and used exclusively for the purposes of this study. Also, rigorous anonymity was kept during the overall data processing and storage. As the members’ activity was often limited to the period when they are taking care about the person with AD, and thus majority of members left or were inactive after their loved one has passed away, it was not possible to contact the group members and obtain their consent. The ethical approval number is (2-2020). More information on our Ethics Committee for Research is available here: https://www.uhk.cz/en/university-of-hradec-kralove/research/ethics-committee-for-research.

## 4. Results

The results are provided in the structure of individual ADLs concepts and the caregivers issues concept and include their both quantitative and qualitative dimensions.

### 4.1. Descriptive Statistics of Individual Categories

The number of posts in individual ADLs categories ranged from 4 (shopping) to 199 (feelings of exhaustion and giving up), on average 60 posts per category were published (mean = 60.29; standard error = 8.68; median = 38; standard deviation = 51.38 and the sample variance 2639.80). Engagement of group members in individual categories was expressed by an average calculated from engagement scores of posts included in a given category. This average score ranged from 23.4 (transferring) to 166.0 (aging, death and dying), average engagement per category reached 69 reactions and comments (mean = 68.57; standard error = 4.78; median = 59; standard deviation = 28.27 and the sample variance 799.19).

### 4.2. Discussion Topics Related to Basic ADLs

#### 4.2.1. Quantitative Dimension

Basic ADLs represented 15.0% out of the total sample. In relation to frequency of communication it is obvious, that more effort is given to activities in which the carer and the person with AD have naturally participated in the past (such as communication and food) than to activities that the person was able to perform independently (such as hygiene, dressing or transferring).

The *interpersonal communication* between the person with AD and other people (mainly caregivers) was the most frequently communicated daily activity in the discussion group; it represented almost half of all comments (46.8%) of Basic ADLs. The topic of *self-feeding* issues of person with AD was recorded in nearly one-fifth of published posts (19.4%).

In relation to engagement of group members, there were no significant differences among categories found. The category of *interpersonal communication* received the highest average engagement rate (79.3). Higher engagement was recorded in the categories of *toilet hygiene* and *self-feeding, eating and drinking* (58.7, resp. 58.6) and *bathing and showering* (57.5). Lower engagement (23.0) was found in the category of *transferring (functional in-house mobility)*. The ANOVA testing confirmed that significant differences among average engagement of discussers in categories of ADLs exist, when *p* = 0.00715451; effect size f was medium (0.250). Tukey’s HSD test showed a statistical difference between the group of transferring and communication. Detailed results of publishing activity as well as engagement in each discussion category is included in Table 1.

#### 4.2.2. Qualitative Dimension

In the most frequently posted category of *communication and interaction*, the caregivers were either mainly making sure themselves knew how to communicate with the person with AD (“What should I tell my mother when she asks why she can’t remember where the dishes go in the kitchen?”; “Should I keep reminding her that she’s forgetting or just pretend we never talked about it?”) or shared about repetitive constant requests of their relatives: “I am no longer able …to recover with the constant calls can you change the channel”; “And here we go… what day is it? Yes…a few times in about 5 min repeating”; “She’s asked me 8 times within 30 min”. They also discussed other questions which are irrational from the perspective of a healthy person: “Dad: “I wonder if you’re still going to grow taller… Do you think you might grow another inch or two yet?” Me: “Dad, I’m 48. I’m pretty sure I’m all done growing.” Also, the experiences with recommended solutions were communicated: “I tried to distract and redirect but she wouldn’t have it. We had a great day together, but sometime after 5 pm, Mr. Hyde decided to take over, even after her pm meds and supplements.” Moreover, having more serious conversation with the person with AD can be hard. “Having a frustrating (yet smiling and ‘nice’) conversation with Dad, trying to convey and discuss some very serious things, but knowing it’s all for nothing.” Besides the mentioned, the caregivers as the close relatives, often miss the conversation itself: “One of the things I really miss about my wife is her voice and conversation. She rarely talks now (stage 6, Alzheimer’s), just whispers the occasional yes and no. Thankfully she still has her beautiful smile.”

The main content of other discussion topics structured according to Basic ADLs is described in detail in Table 2.

### 4.3. Discussion Topics Related to IADLs

#### 4.3.1. Quantitative Dimension

The IADLs included around one-sixth (16.0%) out of the total sample. In relation to publishing frequency, the topics related to the health care and well-being of persons with AD reached the highest communication frequency. The topic of *taking prescribed medications* prevailed in the discussions with more than one-third (38.4%) of all published comments in the category. The topic of *entertainment including movies, music, books, pictures, and video* was recorded in more than one quarter of published posts (25.9%); and the topic *moving within the community* was communicated in nearly one-eighth of posts (12.8%). On the other side, the topics rather associated with activities outside the house as *managing money* (2.4%)*; physical activities, sports* (2.4%)*; using the telephone, computer and online communication* (2.4%); and *shopping* (1.4%) were communicated only rarely.

In relation to the engagement rate, the “having fun” discussion categories of *entertainment* (104.8) and *moving within the community* (86.7) received the highest support from the other discussion group members. Higher engagement was recorded also in categories of *preparing meals* (75.3) and *shopping* (64.5). The ANOVA testing confirmed that significant differences among average engagement of discussers in categories of ADLs exist when *p* = 0.00132005; the effect size f was medium (0.250). Tukey’s HSD test proved a statistical difference between engagement in categories of *taking prescribed medication* and *entertainment* (*p* = 0.00026772), and a weaker difference was found between engagement in categories of *cleaning and maintaining the house* and *entertainment* (*p* = 0.0660254). Detailed results of publishing activity as well as recorded engagement in each discussion category is provided in Table 3.

#### 4.3.2. Qualitative Dimension

In the most communicated category of *taking prescribed medications* the members discussed mainly the effects and experience with particular pills, especially to calm the person with AD: “Curious what your experience has been if your LO has been prescribed Sertraline (Zoloft)?”; “I guess adding the Aricept is helping!!!“—is this norm or should his meds for antidepressants be revaluated?” Besides the traditional medicines, an alternative such as cannabis or CBD oil and their use/non-compliance have also been widely discussed. Many times, the caregivers wanted to assure themselves and raised questions on individual medicines: “Is anyone using CBD for their LO? Outcome?”.

In the second most frequently discussed topic of *entertainment*, its positive effect on the patient’s behaviour was often discussed: “Watching Home Alone with mom, she’s quite enjoying it“; “Music therapy all the way!“; “This works! When I play French music for mama, it takes her back to a happier time; and she hums, sways back and forth and always has a story from her youth at the ready”. Also, the group members asked for recommendations: “Does anyone have suggestions for iPod games? My sister is stage 5/6.”; “I propose that we list those special songs/music of our loved ones.”

Discussion themes of each category of IADLs are described in detail in Table 4.

### 4.4. Discussion Topics Related to EADLs

#### 4.4.1. Quantitative Dimension

Activities of EADLs accounted for the largest part (40.7%) of the research sample. In relation to publishing frequency, there are four discussion topics with a similar representation: *getting help from the family members* (15.3%); *sleeping disorders, sundowning* (14.6%); *appointments; test and surgeries* (14.2%); and *selection of caregivers and health facilities, moving to a facility* (13.6%). On the other side, the themes *care of pets* (2.1%) and *social groups and other public events* (4.1%) are communicated the least.

Obviously, the highest engagement (166.0) was found in category *ageing, death and dying;* this engagement was the highest in all the categories considered in this study. Very high engagement was recorded also in categories *religious observances, holidays, birthdays, anniversaries* (114.5) and *social groups and other public events* (102.2). Vice versa, the lowest engagement was found in categories *care of pets* (45.6) and *safety procedures and emergency response* (43.7).

The ANOVA testing confirmed that significant differences among average engagement of discussers in categories of ADLs exist (*p* < 0.0001); effect size f was large (0.334). Tukey’s HSD test proved statistical difference in many categories: *selection of caregivers and health facilities* and *religious observances, holidays, birthdays and anniversaries* (*p* = 0.0132), *selection of caregivers and health facilities* and *ageing, death and dying* (*p* < 0.0001), *care of pets* and *ageing, death and dying* (*p* = 0.0005858), *safety procedures* and *religious observances, holidays, birthdays and anniversaries* (*p* = 0.0032), *safety procedures* and *ageing, death and dying* (*p* < 0.0001), *sleeping disorders, sundowning* and *religious observances, holidays, birthdays and anniversaries* (*p* = 0.0043), *sleeping disorders, sundowning* and *ageing, death and dying* (*p* < 0.0001), *religious observances, holidays, birthdays and anniversaries* and *getting help from family members* (*p* = 0.0233), *health management, maintenance and prevention* and *ageing, death and dying* (*p* < 0.0001), *ageing, death and dying* and *getting help from family members* (*p* < 0.0001), and *ageing, death and dying* and *appointments, tests, surgeries* (*p* < 0.0001). Detailed results of publishing activity as well as recorded engagement in each discussion category is provided in Table 5.

#### 4.4.2. Qualitative Dimension

There were four highly communicated topics found: getting help from the family, sleeping disorders, medical appointments and tests and selection about the health facility. From more detailed analysis of the posts related to these topics, the probable common denominator turned out to be the decision paralysis on the part of caregivers. Due to the fact of the lack of their time, frequent loneliness and limited professional knowledge about the problem, it is often very difficult for them to decide about another, albeit very close person.

*Getting help from the family members*, the most frequently communicated topic, was characterized, particularly, by sadness that close relatives were not interested in, attending or taking care of the person with AD (“sad day…my sister and brother never even came to visit Mother today…my nieces and nephews never even came to say Merry Christmas to Grandma…and they live right next door. I guess she doesn’t know so it doesn’t matter……”). To a lesser extent, also positive experiences were shared: “Today my sister-in-law did something so wonderful for me. She came to my house and said there was a sale on our favourite ice cream and would I like to go get some? She would stay with Don, my husband, who has dementia. She could’ve just gone and got it for me, but she let me get out and go get it myself. It was so refreshing to get out…” Besides of this, the family controversies and ineffective way of care provided by siblings is described. “ I’m so exhausted even though more people are helping. I’m really looking forward to my brother and his wife moving out.”

The topic of *sleeping disorders, sundowning* was mainly discussed in relation to the person’s inappropriate sleeping schedule affecting the caregiver’s sleep (“*It’s going to be a long night, just put her back to bed for the 3rd time and had to put her depends back on her*.”; “Mom still in bed it’s 2 pm can’t get her up she says no and goes back to sleep?”). The caregivers also shared experiences that can help to calm down the patients (“*Last night after mom had gone to bed I needed to go and sit and hold her hands until she was asleep. Didn’t take long, and you could see her relax and settle in. This is a new phase for her*.”). It is also important to note, that the sundowning issue was found only in 4.69% of posts.

Various kinds of doctor’s visits and medical examinations (EEG, ER, PET scan, CAT scan, mammogram, etc.) have been included in the *appointments, test and surgeries* category. Caregivers also discussed their expectations or experience of a diagnosis of the person with AD and health state development (“Took my Mom to the new neurologist today to get a second opinion. He confirmed that she does have Alzheimer’s. Today my Mom realized that she does have it, and there is no cure. I gave her a hug, told her that my sisters and I love her and will take care of her and not to worry…”), long waiting times for the testing and decisions on surgeries made either by the doctors or themselves.

*Placement of the patient in the health facility* is often described in a larger context and as a more complex life situation. It was usually put in the context with *financial issues* as funding of patient’s care and necessity to sell patient’s house, transfer (or authorization) of POA or insufficient *help of family members*. Discussion themes of each category of EADLs are described in detail in Table 6.

### 4.5. Discussion Topics Related to Caregiver’s Daily Issues

The caregiver’s daily issues (CDIs) concept including 9 categories describing the most often communicated dilemmas and emotions of caregivers was newly proposed based on the content analysis conducted.

#### 4.5.1. Quantitative Dimension

Activities of caregiver’s issues accounted for nearly one-third (30.2%) of the research sample. In relation to publishing frequency, the main topic of *exhaustion, feeling of giving up* was identified as the most frequently published among the caregiver-related activities, in 34.7%. Other topics, a *change of a health state of the patient* recorded in nearly one quarter of published posts (23.4%) and the topic of *violent behaviour* communicated in nearly one-fifth of posts (17.9%), were more frequently discussed by the sampled users. The remaining topics were not communicated significantly often.

The highest engagement (107.3) reached discussion posts related to *caregiver’s success and positive feelings.* However, high engagement (98.4) was also found for the themes *exhaustion and feelings of giving up*. Vice versa, the lowest engagement was found in discussions related to *violent behaviour of the patient* (49.2) and *financial issues* (43.2).

The ANOVA testing confirmed that significant differences among groups (*p* = 0.0002); effect size f was medium (0.221). Tukey’s HSD test proved statistical differences in engagement of the following categories: *change in the state of health of a patient* and *exhaustion, feeling of giving up* (*p* = 0.0062), *exhaustion, feeling of giving up* and *violent behaviour of the patient* (*p* = 0.0020), *exhaustion, feeling of giving up* and *financial issues* (*p* = 0.0197). Detailed results of the published activity as well as recorded engagement in each discussion category is provided in Table 7.

#### 4.5.2. Qualitative Dimension

Generally speaking, and in this concept, the published communication included the need for carers to share their problems and to seek belonging and support. Although some posts were formulated as questions, their nature expected agreement and listening, rather than specifically focused questions. At the same time, the categorization was more difficult, as the individual topics overlapped more than in previous concepts. This fact is also evident from the following detailed characterization of individual categories.

The *exhaustion of caregivers*, the most frequently discussed topic, comprises caregivers complaints about never-ending activity with no breaks, loss of any free time, invisibility of homecare results, feelings of loneliness or not doing enough work, or fear from being more hostile towards their loved one. Caregivers expressed feelings as: “I better get in bed myself to be ready for the next day, but can’t to go because don’t want give up the "My Time”; “Does anyone here resent the person you are caring for because you had to give up your entire life to care for them?” In this context, it is meaningful to note that the fear of the future category contained similar posts, which were bounded with the future of person with AD, caregiver or family state of health: “It is impossible to have next Christmas out of the house, next year.”

Moreover, the desperate situation of the caregivers was caused (or underlined) by the non-improving (or rather worsening) *health state of the patient*. In many cases, the deterioration was also accompanied by unexpected behavioural changes, often as *violent behaviour*. This was related mainly to psychologic violence (insults, cursing), less often also physical violence (throwing of objects, squeezing hand).

In relation to *financial issues* group members published recommendations on how to govern financial matters before the disease starts as shown in the the following statement: “My mother has a three part will. The first two parts are for when she is alive. First is financial POA (requires doctor’s letter). Second is advance directive for medical decisions. Third is the part that splits the assets.” However, the families also experienced critical financial dilemmas: “I don’t know how people survive this disease financially. You make too much money yes though barely make your bills every month so you get no assistance. You basically end up penniless and homeless”; “Alzheimer’s sucks”.

The optimistic experience was summarized in the category caregiver’s success and positive feelings. Caregivers were delighted mainly by positive communication received from the patient (“My mom had a wonderful day. She was feeling so good today”; “Today started with…and thank you’s for caring for me. Just warms my heart.”). The main topics discussed in each category of the CDIs concept are described in detail in Table 8.

## 5. Discussion

This study has classified the statements of family caregivers supporting their loved ones with AD. The findings showed that caregivers communicated more about activities closer to either activities occupying with the persons with AD or about the problems they must cope with. Two-fifths (40.7%) of the communication was related to extended ADLs based on the process of co-occupation between the caregiver and person with AD; almost one-third (30.2%) of the communication was dealing with caregivers issues as dilemmas, fears, or emotions included in the CDIs concept; the other two concepts were not communicated as frequently: instrumentally, ADLs were communicated in 16.0% and basic ADLs in 15.1%.

More detailed insight into the individual categories exposed that the highest need for communication existed in the field of feelings of exhaustion and giving up. The frequency of this category was significantly higher than in other ones and accounted for 200 posts (9.5% from the research sample). There were other frequently shared categories such as interpersonal communication between the caregiver and the person with AD, help from other family members, change in a person’s health state or sleeping disorders and sleeping regime. The highest support of from the group members—expressed by the post engagement score—obtained communication oriented toward *aging, death and dying* and then, with certain distance, on more funny life situations such as celebrations of *birthdays and anniversaries* and the category of *entertainment*.

The summarized picture of the communication frequency of individual categories and the engagement (support) of group members is provided in Figure 2 (categories with a frequency lower than 1% from the total sample were not included).

In general, the study findings confirmed the high level of distress and overall caregiver’ burden presented in many previous studies [70,71,72,73,74] and particularly related to wives and daughters [36]. Specifically, the need to “ventilate” and look for the support was embodied in *the give up feelings* of caregivers. Such online venting as a coping strategy is, according to Lagervall et al. [56], associated with higher caregiver burden and is similar like strategies used in conventional communication in-person support groups. The study also confirmed lack of time for the carers themselves and enabled an insight into the level of experienced stress in comparison to other caregiving activities/issues. The highly communicated topic of *interpersonal communication* is consistent with the study of van Hoof, Verbeek and Janssen [75], where the aspects of effective communication, empathy development and conflict resolution are stressed out. The topic of *getting help from the family members,* on one hand, stressed out the significance of planned interventions focusing on the whole family [76] and barriers existing among the professionals, caregivers and the rest of the family [77]. On the other hand, the caregivers included in this study also communicated the need of regular and on-time re-arrangement and re-assessment of caregiving roles and responsibilities within the family care. The field of *violent behaviour* of people with AD can be considered as an unexpected result of the study. Although Cheng [29] in her study pointed out that disruptive behaviour was more disturbing because of the adverse impact on the emotional connection between the caregiver and care-recipient, the frequency of communication and particularly engagement rate of other members was rather only average.

The results of the study should be interpreted also with consideration of its limitations. Primarily two complex areas should be considered: the first one is related to the caregiver’s behaviour on the social media and social media as such, the second one is related to the research sample structure. Although the main disadvantages of social media, such as the presence of spam and irrelevant or funny communication, were reduced by our sampling strategy and careful manual coding, some potential risks still exist. Users may attempt to receive appreciation or higher support from others by conforming their content of conversation to “trendy” topics or use of exaggeration to look better in front of the others. Potentially, due to the rapid development of social networks, emergence of more technically demanding or locally focused online discussions will make it impossible to replicate this analysis in the future. The research sample limitation may be caused by the fact that the carers included in the sample were primarily citizens of the United States and to a lesser extent the United Kingdom. For this reason, the lessons learned may be difficult to apply in other countries. Also, the high number of daughters included and the low number of spouses in the role of carers included in the sample may distort the knowledge gained. Secondly, the fact that the article has only one author, on the one hand, can increase in the processing and presentation of findings, on the other hand, can increase the risk of bias or unilateral interpretations.

## 6. Conclusions and Implications

The longitudinal monitoring of communication published by caregivers identified its quantitative and qualitative nature in 26 categories of ADLs concepts and 9 categories of CDIs concept. As a longitudinal approach was taken, we expect that the results reflect the real-life issues of the individual caregivers.

Therefore, such complex insight into the family health-care process and especially caregiver’s burden enables: (1) to obtain a comprehensive picture of demands of the work of family carers, particularly in the field of psychological, emotional, physical, social and also time and financial demands; (2) to get better understanding how the caregiving process affects family relationships and family time, care or even financial arrangement; (3) to find out that some caregiving activities are more recognized in the caregiving community than others; (4) based on the preceding points we will be more able to set up at least minimal measurable objectives and metrics to measure caregivers activity that are necessary for caregivers to reassure them that they are effective and beneficial; and finally (5) better understanding of issues related to local and national health policies towards family care. In general, the study findings revealed in detail more aspects of QoL of the caregivers of people with AD. Also, from the methodological perspective, the conducted analysis of caregivers’ statements based on long-term monitoring of social media discussion group can become an important platform for better understanding of real caregiver’s life situations.

The caregiver burden can be reduced by interventions recommended interventions of carers focusing on psychoeducation, general support provided by an informal support group, respite (planned, temporary relief for caregivers), cognitive behavioural therapy, counselling in the field of family conflict and more activities [9,74,78]. However, the study findings can significantly support and enrich previous studies in relation to the found structure of mostly challenging daily activities and the nature of caregivers’ issues, questions and their emotions.

There are many future research opportunities in the area. Social media brings a wealth of quantitative and qualitative data, which, if examined in a proper methodical way, can bring new and enriching knowledge documenting the life of carers of people with AD. Therefore, future research studies can be oriented, for example, on in-depth analysis of individual daily living activities, use of more advanced methods of text-mining, text and sentiment analysis, and others.

## Figures and Tables

**Figure 1 ijerph-17-04412-f001:**
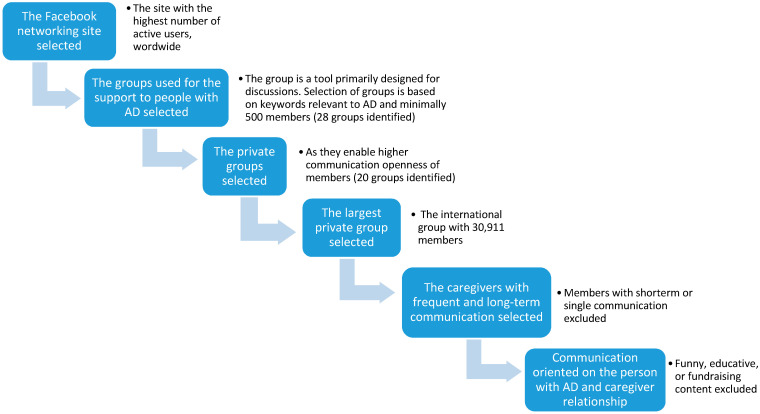
Individual stages of sampling strategy.

**Figure 2 ijerph-17-04412-f002:**
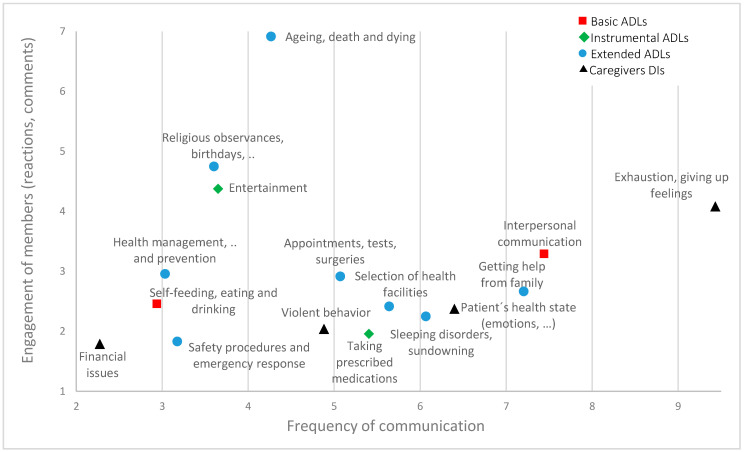
Most frequently communicated topics according to categories and their engagement (in %).

**Table 1 ijerph-17-04412-t001:** Communication of group members in relation to basic activities of daily living (ADLs), *(n* = 317).

Category of ADLs	Posts		Average Reactions per Postt	Average Comments per Post	Average Total Engagement per Post
	Absolute	%			
Bathing and showering	19	5.8	43.5	14.3	57.8
Personal hygiene	18	5.5	38.2	10.8	49.0
Dressing	15	4.6	30.3	18.3	48.6
Toilet hygiene	30	9.2	27.7	30.9	58.6
Transferring ^1^	20	6.2	12	11.4	23.4
Self-feeding, eating/drinking ^2^	63	19.4	34.6	24.2	58.8
Communication and interaction	152	46.8	59.3	19.6	78.9
In total for basic ADLs	319	15.1 ^3^	35.0	18.4	71.9

Note: ^1^ Functional mobility within the house; ^2^ including chewing and swallowing; ^3^ out of the total research sample.

**Table 2 ijerph-17-04412-t002:** Communication of group members in relation to basic activities of daily living (ADLs).

Category of ADLs	Main Topics Discussed in the Category
Bathing and showering	Stress reducing toys during hair wash, refusing to wash (the whole body or a specific part), misconception of person with AD that they were already washed, positive feelings after washing, how to inform about it, recommended methods, fear of reactions when being told about washing.
Personal hygiene	Dental hygiene, new haircut, shaving, beard care.
Dressing	Struggles when putting underwear (particularly bra), dressing inappropriate for the weather, comfortable clothes, undressing.
Toilet hygiene	Stressfulness when diapers are rejected, control of bowel movement, diarrhoea and constipation, night incontinence, excretion outside the toilet, frequent change of bedding, physical difficulty of changing diapers.
Transferring ^1^	Injuries caused by transfer, rejection of the use walkers or other mobility aids and its consequences, visual impairment.
Self-feeding, eating/drinking ^2^	Insufficient diet, loss of appetite, over drinking (coffee, coke), overeating (in general, sugar), rejection of solid food, swallowing issues.
Communication and interaction with the patient	Missing of close voice, conversation, loss of sense of humour, loss of orientation in time, short-term return of conversation ability and answering, sighing/examples of nonsensical/funny conversations, informing what we plan to do in the future.

Note: ^1^ Functional mobility within the house; ^2^ including chewing and swallowing.

**Table 3 ijerph-17-04412-t003:** Communication of group members in relation to instrumental activities of daily living (IADLs) (*n* = 297).

Category of IADLs	Posts		Average Reactions per Post	Average Comments per Post	Average Total Engagement per Post
	Absolute	%			
Cleaning and maintaining the house	22	7.4	22.5	13.5	36
Managing money	7	2.4	23.1	21.3	44.4
Moving within the community ^1^	38	12.8	69.7	16.5	86.2
Preparing meals	21	7.1	32.3	42.7	75
Shopping	4	1.3	35.5	28.5	64
Taking prescribed medications	114	38.4	18.2	28.8	47
Using the phone, computer and online communication	7	2.4	28.1	16.7	44.8
Entertainment ^2^	77	25.9	82.8	21.8	104.6
Physical activities, sports	7	2.4	24.3	17.4	41.7
In total for IADLs	297	16.0 ^3^	37.4	23.1	83.7

Note: ^1^ Outside the house and/or family. ^2^ Including movies, music, books, pictures, videos. ^3^ Out of the total sample.

**Table 4 ijerph-17-04412-t004:** Qualitative description of the main topics related to instrumental activities of daily living (IADLs).

Category of Daily Life Activity	Main Topics Discussed in the Category
Cleaning and maintaining the house	Lack of self-cleaning, cleaning in a confused way (storing of objects in strange places, mismatching of socks), unwillingness to get rid of objects, wrong time for cleaning.
Managing money	Bank account transfer on other persons, accusation by the person with AD from stealing money.
Moving within the community ^1^	Visiting local shops, cafes, museums, manicure, games, etc., and its impact on the behaviour, common eating outside the home, trips, and holidays (from one to several days), discussions on ability to drive.
Preparing meals	Composition and nutritional value of food, variety of dishes, food quality verification, recommendations for new “soft” or liquid/puree food (in later stages of disease); providing a varied diet, selection of machine to puree foods, recipes for good foods and drinks.
Shopping	Caregiver’s stress when shopping together (watching many things at once), loss of place orientation when shopping.
Taking prescribed medications	Effects (including side effects) of prescribed medication (Memantime, Zoloft, Risperidone, Donepezil, Xanax, Aricept), risks of specific alternative drugs (mainly CBD oil, cannabis), change of medication and its impact, troubles with swallowing medicine.
Using the phone, computer and online communication	Excessive costs caused by calls, discussions on virtual assistants (Alexa), advantage of Skype for distant conversation, repetitive, repeated calls (50 times a day) or nonsensical calls from the person with AD.
Entertainment ^2^	Positive effects of music listening on a patient, use of headphones, share of selfies/videos with patient, cinema visits, video recording together, recommendations on good movies, playing music instrument by person with AD, family picture time, iPod games.
Physical activities, sports	Surprise over good sport performance of the person with AD, resistance to return to sports.

Note: ^1^ outside the house and/or family. ^2^ includes movies, music, books, pictures, videos.

**Table 5 ijerph-17-04412-t005:** Communication of group members in relation to extended activities of daily living (EADLs) (*n* = 858).

Category of EADLs	Posts		Average Reactions per Post	Average Comments per Post	Average Total Engagement per Post
	Absolute	%			
Selection of caregivers and health facilities, moving to a facility	119	13.6	35.0	23.4	58.4
Care of pets	18	2.1	33.8	11.8	45.6
Safety procedures and emergency responses	67	7.6	25.9	17.8	43.7
Sleeping disorders, sundowning	128	14.6	36.1	18.5	54.6
Religious observances, holidays, birthdays, anniversaries	78	8.9	93.1	22.3	115.4
Health management, maintenance and prevention	64	7.3	39.8	31.0	70.8
Social groups and other public events	36	4.1	80.2	22.0	102.2
Ageing, death and dying	90	10.3	116.6	49.4	166.0
Getting help ^1^	134	15.3	43.2	23.9	67.1
Appointments, tests, surgeries	124	14.5	35.1	28.7	63.8
In total for instrumental ADLs	858	40.7 ^2^	54.0	24.9	103.8

Note: ^1^ from family members or outside the family (nurses); ^2^ out of the research sample.

**Table 6 ijerph-17-04412-t006:** Qualitative description of the main topics related to extended activities of daily living (EADLs).

Category of EADLs	Main Topics Discussed in the Category
Selection of caregivers and health facilities, moving to a facility	Right time to place the person to facility, emotions/guilt associated with leaving to the facility (for both sides), selection of appropriate long-term/day care facility, waiting time for admission to the facility, health assessment/qualification for the level of care needed, change of the facility, beginnings in a new facility.
Care of pets	Significance of animals for patients, missing animals during their stay in a facility, animal death and its effect on the patient.
Safety procedures and emergency responses	Selecting bracelets/watches with GPS tracking (user friendly for patients), remote monitoring of physical activity, and coordination during use of a walker. Leaving a patient at home alone, patients’ escapes, concerns of caregivers when the person starts to walk again after a longer period on the bed, concerns related do the use of care and keeping car keys/driving license. Unexpected falls and its prevention. Safety in bathroom.
Sleeping disorders, sundowning	Persons wake up at night and start a daily routine; persons are several days without sleeping, effort to respect a sleeping schedule; concerns and discussions over “sleeping” medication. In lesser extent the sundowning issues and effects.
Religious observances, holidays, birthdays, anniversaries	Birthdays, Mother’s and Father’s Days, Marriage anniversaries, graduations in the family and other celebrations on the side of patient/caregiver/family. Positive feelings and behaviour of patients, effort to include patient in the selection of gifts. Celebrations with grandchildren.
Health management, maintenance and prevention	Management of caring activities and transfer of some works on other family members; concerns about the disease inheritance. Recommendations how to “cure”. How to recognize dementia/AD and diagnostics of disease. Efforts to increase patient independence.
Social groups and other public events	Activity groups, adult-day camps, activities of local Alzheimer’s organizations, fundraising events (walk/run)
Ageing, death and dying	Asking for support in hard, life threatening situations for patients. Discussions on “how much time do we have left”. Description of last life moments. Dilemma how to tell the other about the patient’s death. Memories on and love to the person who died. The joy of abandoning a miserable quality of life, asking for prayers and support.
Getting help ^1^	Family disputes, controversies and no communication from the side of siblings/other relatives. Ineffectiveness of help from certain people living close. Help of other family members and its positive effect on the caregiver/person with AD. Help from the hospices and other (non-relatives) persons.
Appointments, tests, surgeries	Appointments to recognize patient’s needs, medication. Hospital tests for the disease stage. Surgeries after patient’s falls or injuries. Pros and cons of testing for AD predisposition.

Note: ^1^ from family members or outside the family (nurses).

**Table 7 ijerph-17-04412-t007:** Communication of group members in relation to caregiver´s daily issues (CDIs) (*n* = 638).

Category of CDIs	Posts		Average Reactions per Post	Average Comments per Post	Average Total Engagement per Post
	Absolute	%			
Quality of health services	18	3.1	43.8	42.7	86.5
Change of a health state of the patient	135	23.4	33.6	22.8	56.4
Physical recognition of close relatives	42	7.3	63.0	21.7	84.7
Exhaustion, feeling of giving up, guilt	200	34.7	59.4	39	98.4
Fear of the future	33	5.7	47.5	38.4	85.9
Violent behaviour of the patient towards to caregiver	103	17.9	28.2	21	49.2
Caregiver’s success and positive feelings	26	4.5	96.2	11.1	107.3
Financial issues	48	8.3	20.9	22.3	43.2
Discussion group support and others	33	5.7	42.9	37	79.9
In total for CDIs	638	30.2 ^1^	48.4	28.4	76.8

Note: ^1^ Out of the total research sample.

**Table 8 ijerph-17-04412-t008:** Qualitative description of the main topics related to CDIs.

Category of Daily Life Activity	Main Topics Discussed in the Category
Quality of health services	Assessment of certified nurse assistance (CNA) quality, patient’s rejection of external services, complains on hospital services and its communication about the patient’s health state.
Change of a health state of the patient	Entering a new stage of the disease and coping with this new situation; associated emotions of caregivers. Patients (usually parents in the past) are becoming to be like children and children like parents. Description of a new specific health issue of the patient: physical (dry skin, rash, bedsores) and mental ones (agitation, frustration, murmuring, complaining teeth grinding, walking back and forth in the house).
Physical recognition of close relatives	Temporary or total non-recognition of a close person (life partner and children), forgetting the name. Comments on disappointment and other emotions associated. Fear of caregivers (often children) that they will not be recognized.
Exhaustion, feeling of giving up, guilt	Effects of caregiving activity performed 24/7 with no breaks, health issues of caregivers. Potential caregivers’ hostility towards the patient. Caregivers’ impossibility of having their “my time”. Loneliness. Guilt coming from not sufficient/successful work of caregivers (patient’s health state is not getting better).
Fear of the future	What the situation will look like in the future if the current situation is hard? Losing of personal life and time (relationships, hobbies, and friends) because of caregiving and fear of the future. Concerns about the future health state of the person with AD.
Violent behavior of the person with AD towards to the caregiver	Patient’s psychical (insults, cursing, shouting) and physical violence (hitting, throwing objects). The person with AD is rude when the caregiver wants to leave. Unexpected turns of person’s behaviour.
Caregiver’s success and positive feelings	Person’s thanks and love expression towards the caregivers. Finding pleasure in nature of caregiving work. Pleasure from the gifts received from the person with AD.
Financial issues	Absence/presence of medical/financial POA and/or patient’s will, communication between the person with AD and family about financial affairs, costs of health facilities and what is/is not covered by state medical care, paying of home health providers/caregivers, future concerns on financial matters. Accusations from patients/relatives about disuse of money; quarrel about money. Difficult financial family situations, tips on the ways of fundraising (for example crowdfunding) and proposals how to change a health insurance system.
Group support	Appreciation for group membership and support; opportunity to express emotions; get advice; resolve the situation; communicate with people who have the same problem, searching for someone in the group physically living nearby.

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
