# Peer review of "Caregivers’ Experience of Caring for a Family Member with Alzheimer’s Disease: A Content Analysis of Longitudinal Social Media Communication"

_ijerph, 2020, doi:10.3390/ijerph17124412_

Round 1
Reviewer 1 Report
References cited as specific to social media and dementia caregiving are studies that do not solely focus on this population. More recent studies in the field that are specifically focused on this population are not referenced or discussed in terms of background or explaining how the findings of the current study align with the extant literature.
The rationale for supporting the study purpose (i.e., using Facebook to explore how caregivers manage daily activities) remains unclear in terms of why use this discussion group as a data source.
How were the 2110 posts from the 100 Facebook group members chosen? Did this constitute all of the posts from these individuals over the period of time indicated? Or were posts chosen based on some other criteria? Were comments to posts included in the analysis?
I’m not sure that I would describe high frequencies of posts in a particular topic as engagement. This term suggests that participants in the sample were interacting in some way (e.g., commenting on these posts). Given that it does not appear that comments to posts were analyzed, overstatement of the type of data and results should be avoided.
The Discussion section of the manuscript remains very sparse. There is no discussion of how the results compare or contrast with results from other research examining the use of social media by this study population specifically.
Author Response
Dear reviewer, the answers are available in the text below.
References cited as specific to social media and dementia caregiving are studies that do not solely focus on this population. More recent studies in the field that are specifically focused on this population are not referenced or discussed in terms of background or explaining how the findings of the current study align with the extant literature.
Author: More recent studies focusing on the relevant population were added. Also, this part was re-structured and I hope it is more logical and consistent, at the moment.
The rationale for supporting the study purpose (i.e., using Facebook to explore how caregivers manage daily activities) remains unclear in terms of why use this discussion group as a data source.
Author: This rationale was added and is available in lines 172-177.
How were the 2110 posts from the 100 Facebook group members chosen? Did this constitute all of the posts from these individuals over the period of time indicated? Or were posts chosen based on some other criteria? Were comments to posts included in the analysis?
Author: At the first step, the individuals with longer publishing history posts (at least 10 posts and half a year history) were collected. At the next step, the posts from the individuals that are not directly oriented on the relationship between the caregiver and person with AD were excluded from the analysis. Excluded posts contained mainly funny, educative or fundraising content.
Comments were included in the analysis only as a part of engagement rate (sum of likes and comments). This sampling strategy is described in Figure 2 and the relevant text was revised in the paper (lines 203-212).
I’m not sure that I would describe high frequencies of posts in a particular topic as engagement. This term suggests that participants in the sample were interacting in some way (e.g., commenting on these posts). Given that it does not appear that comments to posts were analyzed, overstatement of the type of data and results should be avoided.
Author: There are two main findings: (1) frequency of the published posts categorized according to content, and (2) the level of engagement of these posts. The engagement is an established term for measuring the feedback of other users of social media. An engagement rate represents a total sum of likes, comments and sharings (described in lines 162-171). Thus the engagement was also measured by this way, and it provides information about the support of other group members, respectively how the certain type of topic was accepted/supported by the other discussants.
The Discussion section of the manuscript remains very sparse. There is no discussion of how the results compare or contrast with results from other research examining the use of social media by this study population specifically.
Author: The section was revised, new literature sources to compare/contrast results used.
Reviewer 2 Report
I am agree with author's response.
Author Response
Thank you very much for your positive feedback.
Reviewer 3 Report
It was a pleasure to read your revised manuscript. Thank you for the careful revisions and willingness to do so. The paper is interesting, well written and structured, and original, and the findings are evocative. From my experience with people who are caregivers of someone with dementia, the findings also ring true. I believe the study will contribute to better understanding of the many challenges that carers face.
I have just a few comments and suggestions for final refinements to the manuscript, as follows:
- The title still does not reflect the study design. My suggestion is 'Caregivers' experience of caring for a family member with Alzheimer's Disease: A content analysis of longitudinal social media postings', or something like that.
- The last sentence of the abstract (the conclusion) does not make sense: suggest revise this.
- Regarding your access to the social media group, how was this obtained? As the group was restricted to people who knew a person affected by a neurocognitive disorder. Also, you give an excellent summary of the ethical implications of using social media content for research purposes, but there was little information about the precise ethical implications of your study. While you state the study was approved by an ethics committee, I think it is important to articulate the specific considerations and how you addressed or justified these.
- Figure 3: are the colours in the key meant to match the colours in the figure?
- Lines 511-13: suggest re-write to say that those behaviours MAY have occurred.
- Regarding being the solo researcher/author on the study, I think this is another limitation that should also be explicitly acknowledged. Having several researchers on a study helps to increase study rigor by reducing the risk of bias, which each one of us has. Our colleagues help us to see our own biases, other ways of seeing the data, and to ensure a circumspect interpretation. While I believe your findings and interpretations to be trustworthy, being a solo researcher/author is an important limitation to note.
- Suggest a final overall edit/check, as there are still several points where present tense was incorrectly used.
Author Response
Dear reviewer, my response is included in the text of your review, below.
It was a pleasure to read your revised manuscript. Thank you for the careful revisions and willingness to do so. The paper is interesting, well written and structured, and original, and the findings are evocative. From my experience with people who are caregivers of someone with dementia, the findings also ring true. I believe the study will contribute to better understanding of the many challenges that carers face.
Author: Thank you very much for your useful review and support, it is not so usual.
I have just a few comments and suggestions for final refinements to the manuscript, as follows:
- The title still does not reflect the study design. My suggestion is 'Caregivers' experience of caring for a family member with Alzheimer's Disease: A content analysis of longitudinal social media postings', or something like that.
Author: The title was changed according to your suggestion.
2. The last sentence of the abstract (the conclusion) does not make sense: suggest revise this.
Author: This conclusion of the abstract was changed.
3. Regarding your access to the social media group, how was this obtained? As the group was restricted to people who knew a person affected by a neurocognitive disorder. Also, you give an excellent summary of the ethical implications of using social media content for research purposes, but there was little information about the precise ethical implications of your study. While you state the study was approved by an ethics committee, I think it is important to articulate the specific considerations and how you addressed or justified these.
Author: This part was revised and now is available in lines 274 to 281.
During the entering the group I have answered that I know such person (and it is a truth).
4. Figure 3: are the colours in the key meant to match the colours in the figure?
Author: The Fig. 3 was revised, and color has been added to the symbols in the key
5. Lines 511-13: suggest re-write to say that those behaviours MAY have occurred.
Author: This part was re-written according to your recommendation.
6. Regarding being the solo researcher/author on the study, I think this is another limitation that should also be explicitly acknowledged. Having several researchers on a study helps to increase study rigor by reducing the risk of bias, which each one of us has. Our colleagues help us to see our own biases, other ways of seeing the data, and to ensure a circumspect interpretation. While I believe your findings and interpretations to be trustworthy, being a solo researcher/author is an important limitation to note.
Author: Thank you for your recommendation and I agree with your opinion. „The solo authorship“ was included among the limitations of the study.
7. Suggest a final overall edit/check, as there are still several points where present tense was incorrectly used.
Author: The study was checked by an English language professional, so I hope it helped.
Round 2
Reviewer 1 Report
The authors have addressed the comments.
This manuscript is a resubmission of an earlier submission. The following is a list of the peer review reports and author responses from that submission.
Round 1
Reviewer 1 Report
It is a misstatement to say that the research on ADLs in dementia caregiving focuses on pharmaceuticals and natural products. Such a statement discounts a great deal of the research related to nonpharmacological approaches to address symptom management and caregiver burden, which makes up the majority of literature in this space.
The introduction does not describe a clear rationale for the study in terms of both the focus on activities of daily living and the use of social media as a data source.
It is not clear if the list of extended and caregiver ADLs were derived from the extant literature or from the data analysis process. What is described as caregiver ADLs do not appear to be ADLs at all but rather as responses to caregiving, mostly emotional.
Were members of the Facebook group informed of the research? Were they provided with an option to opt out of the study? The ethical approach to the study and whether it was approved by an ethics committee is not described.
Were Facebook data collected prospectively or retrospectively? If prospectively, how did the presence and monitoring of the group by the research influence the posts from caregivers and people with dementia?
Were individual posts coded with more than one category? The qualitative methodology used in the analysis is not described.
In the abstract, the sample was described as dyads. However, it does not appear that true dyads were analyzed. Rather the data included posts from caregivers and people with dementia.
The discussion section does not situate the findings of the analysis within the existing literature, particularly previous studies examining social media data from caregivers of people with dementia. The limitations and strengths of the study are not described.
Finally, the manuscript needs to edited for clarity and grammar.
Author Response
Dear reviewer,
R: It is a misstatement to say that the research on ADLs in dementia caregiving focuses on pharmaceuticals and natural products. Such a statement discounts a great deal of the research related to nonpharmacological approaches to address symptom management and caregiver burden, which makes up the majority of literature in this space.
A: Thank you very much for your comment. The text was revised and more attention to caregiver burden was paid.
R: The introduction does not describe a clear rationale for the study in terms of both the focus on activities of daily living and the use of social media as a data source.
A: The rationale and interconnection between ADLs and social media use were revised, now it is available in lines 57-67.
R: It is not clear if the list of extended and caregiver ADLs were derived from the extant literature or from the data analysis process. What is described as caregiver ADLs do not appear to be ADLs at all but rather as responses to caregiving, mostly emotional.
A: This part was revised and corrected in the text. ADLs concepts serve as the prescribed coding scheme, caregiver ADLs concept was renamed (since they are not real activities) and it was described that it was derived from the data collected.
R: Were members of the Facebook group informed of the research? Were they provided with an option to opt out of the study? The ethical approach to the study and whether it was approved by an ethics committee is not described.
A: The study was approved by an ethics committee. This part was revised and I hope better formulated; now it is available in section 2.4.1 Data collection.
R: Were Facebook data collected prospectively or retrospectively? If prospectively, how did the presence and monitoring of the group by the research influence the posts from caregivers and people with dementia?
A: The data were collected retrospectively.
R: Were individual posts coded with more than one category? The qualitative methodology used in the analysis is not described.
A: The individual posts were assigned to just one category according to the prevailing content. This fact is now described in the text (section 2.4.1, line 218).
R: In the abstract, the sample was described as dyads. However, it does not appear that true dyads were analyzed. Rather the data included posts from caregivers and people with dementia.
A: Analysis was based on the longitudinal monitoring of communication published by the caregivers, but containing only communication towards the specific person with AD. Another communication related to the general issues of life, relationship with other persons, or disease was excluded from the posts analyzed. For this reason, these relationships were called dyads and I still believe that they can be called like this.
R: The discussion section does not situate the findings of the analysis within the existing literature, particularly previous studies examining social media data from caregivers of people with dementia. The limitations and strengths of the study are not described.
A: The section of discussion was completely revised; several new studies used to discuss study findings. Also, the limitations and strengths of the study were described newly in the section.
Finally, the manuscript needs to edit for clarity and grammar.
The text was revised, and I hope it is of better quality, also because of the effort of the reviewers. Thank you very much for your remarks.

Reviewer 2 Report
Thank you for the opportunity to review the manuscript entitled "Activities of Daily Living of Patients with Alzheimer´s Disease. Social Media Evidence by Family Caregivers” for International Journal of Environmental Research and Public Health. The core finding was examined and revealed the structure of caregivers’ experiences, issues and emotions related to activities of daily living and its acceptance of peer caregivers. This finding is meaningful and represent an important area of inquiry that is relevant to the readership of this journal. This finding is meaningful and represents an important area of inquiry that is relevant to the readership of this journal. The statistical analyses were well conducted. However, additional information such as standard deviation should be provided in tables to improve the manuscript.
Author Response
Thank you for the opportunity to review the manuscript entitled "Activities of Daily Living of Patients with Alzheimer´s Disease. Social Media Evidence by Family Caregivers” for International Journal of Environmental Research and Public Health. The core finding was examined and revealed the structure of caregivers’ experiences, issues and emotions related to activities of daily living and its acceptance of peer caregivers. This finding is meaningful and represent an important area of inquiry that is relevant to the readership of this journal. This finding is meaningful and represents an important area of inquiry that is relevant to the readership of this journal. The statistical analyses were well conducted. However, additional information such as standard deviation should be provided in tables to improve the manuscript.
Thank you very much for your positive feedback. The standard deviations are available in subchapter 2.3.2. I agree with your opinion that it would be better to present standard deviations in tables. However, as the study is quite long now, I am afraid that another table would make the study more complicated and less comfortable to read.
Reviewer 3 Report
It is a complex study, with an extremely relevant importance in the social sciences and health care.Well-structured study with a quantitative and qualitative assessment, with an appropriate methodology, and with the limitations mentioned by the authors and which may lead to future investigations.
The evaluation between coders showed a high reliability, translating evaluation criteria and rigor among researchers.
Author Response
R: It is a complex study, with an extremely relevant importance in the social sciences and health care.
Well-structured study with a quantitative and qualitative assessment, with an appropriate methodology, and with the limitations mentioned by the authors and which may lead to future investigations.
The evaluation between coders showed a high reliability, translating evaluation criteria and rigor among researchers.
A: Thank you very much for your positive feedback.
Reviewer 4 Report
Thank you for the opportunity to review your manuscripts, ‘Activities of Daily Living of Patients with Alzheimer´s Disease. Social Media Evidence by Family Caregivers’.
This is an interesting paper and begins promisingly. The topic is a worthy one, and the approach to data collection was quite novel.
However, there are some areas requiring revision before the manuscript is ready for publication. These are as follows:
- Title should be one sentence, not two. Consider replacing the full stop with a colon.
- The title refers only to the ADLs of persons with Alzheimer’s, whereas the study examined ADLs also of the caregivers.
- Please re-consider the use of the word ‘patients’ for the persons with Alzheimer’s, as this is not a term people with dementia like to be called outside of a medical context. E.g. see Dementia Language Guidelines: https://www.dementia.org.au/files/resources/dementia-language-guidelines.pdf
- The title should include the study design
- Introduction: Lines 61-68 are not customary or needed.
- Section: ‘Theoretical Background’ is wrongly named, as the discussion on social media research is not theory, but background information. Nor are ADLs a theory; instead they are a construct or classification system.
- The addition of ADLs for family caregivers for the purposes of this study is highly problematic. Firstly, none of the listed categories are activities. Secondly, how where these categories derived? If they were derived from the data collected in this study, then they are findings, not pre-defined categories. This part of the study requires re-considering, or better explanation/justification.
- It seems that people observed on the Facebook group were not aware that they were being observed for the purposes of research, and therefore did not give their consent. The ethical implications of collecting data from social media for the purposes of research should be clearly stated, and a rationale given why it was not possible to seek people’s consent for their posts to be used.
- Figure 2 does not make sense to me – please consider revising this for greater clarity.
- Lines 194-200 are results, not methods.
- Analysis section: The description of the quantitative and qualitative analysis is not sufficient. E.g. there is no description of the statistical methods that were used, and qualitative analysis is more than giving quotes according to category. If content analysis was used (as was stated earlier), what are the normative process of this method?
- Results: This is the section of the manuscript is the most problematic. While I got a sense of the highs and lows of peoples’ lives, results are simply and (too lengthily) presented as categories of ADLs. A higher order of interpretation is required here.
- It is unusual these days for a research manuscript to have only one author. Why wasn’t the student involved in data collection and interpretation included as a co-author?
- The manuscript requires revisions to ensure better English expression, correction of grammatical errors (including those of tense), and that it be made more concise.
I wish you well in the revision of your interesting paper.
Author Response
Dear reviewer,
the point-by-point response is provided below.
R: Thank you for the opportunity to review your manuscripts, ‘Activities of Daily Living of Patients with Alzheimer´s Disease. Social Media Evidence by Family Caregivers’.
This is an interesting paper and begins promisingly. The topic is a worthy one, and the approach to data collection was quite novel.
However, there are some areas requiring revision before the manuscript is ready for publication. These are as follows:
- Title should be one sentence, not two. Consider replacing the full stop with a colon.
- The title refers only to the ADLs of persons with Alzheimer’s, whereas the study examined ADLs also of the caregivers.
A: Thank you very much for your fruitful comments. The title of the study was revised and changed.
R: Please re-consider the use of the word ‘patients’ for the persons with Alzheimer’s, as this is not a term people with dementia like to be called outside of a medical context. E.g. see Dementia Language Guidelines: https://www.dementia.org.au/files/resources/dementia-language-guidelines.pdf
A: The word „patient“ was substituted by the word people or person wu in the text, where it was related to people with AD.
R: The title should include the study design
A: The title of the study was revised and changed.
R: Introduction: Lines 61-68 are not customary or needed.
A: Lines 61-68 were removed.
R: Section: ‘Theoretical Background’ is wrongly named, as the discussion on social media research is not theory, but background information. Nor are ADLs a theory; instead they are a construct or classification system.
A: The section was revised, including the title of the section.
R: The addition of ADLs for family caregivers for the purposes of this study is highly problematic. Firstly, none of the listed categories are activities. Secondly, how where these categories derived? If they were derived from the data collected in this study, then they are findings, not pre-defined categories. This part of the study requires re-considering, or better explanation/justification.
A: This part was revised. The ADLs were used as a coding scheme, the design of caregiver issues of daily living was included in the results. It was better explained in the chapter of methods, the ADLs concepts were better cited.
R: It seems that people observed on the Facebook group were not aware that they were being observed for the purposes of research, and therefore did not give their consent. The ethical implications of collecting data from social media for the purposes of research should be clearly stated, and a rationale given why it was not possible to seek people’s consent for their posts to be used.
A: The study was approved by an ethics committee. The relevant text was revised and is available in the subchapter 2.4.1 Data collection.
R: Figure 2 does not make sense to me – please consider revising this for greater clarity.
A: Fig. 2 was completely reworked to reach higher clarity.
R: Lines 194-200 are results, not methods.
A: The relevant text has been moved to the chapter of Results.
R: Analysis section: The description of the quantitative and qualitative analysis is not sufficient. E.g. there is no description of the statistical methods that were used, and qualitative analysis is more than giving quotes according to category. If content analysis was used (as was stated earlier), what are the normative process of this method?
A: The description of the methods was revised and extended; now includes more information on the procedure of content analysis done.
R: Results: This is the section of the manuscript is the most problematic. While I got a sense of the highs and lows of peoples’ lives, results are simply and (too lengthily) presented as categories of ADLs. A higher order of interpretation is required here.
A: The chapter of results was revised, the prevailing content of the discussion in individual concepts was identified, some examples of caregivers’ statements were shortened to reach higher clarity.
R: It is unusual these days for a research manuscript to have only one author. Why wasn’t the student involved in data collection and interpretation included as a co-author?
A: The primary reason is that my colleague (the student) participated only in the technical classification of the data gathered; the student was not involved in interpretation.
R: The manuscript requires revisions to ensure better English expression, correction of grammatical errors (including those of tense), and that it be made more concise.
I wish you well in the revision of your interesting paper.
A: Thank you very much for your helpful review which, I believe, improved the quality of the study.